# Parents’ Initiation of Alcohol Drinking among Elementary and Kindergarten Students

**DOI:** 10.3390/children8040258

**Published:** 2021-03-27

**Authors:** Kazuko Eto, Masahiro Sugimoto

**Affiliations:** 1Department of Nursing, Yokohama Soei University, Yokoyama 226-0015, Japan; etokaguko@yahoo.co.jp; 2Research and Development Center for Minimally Invasive Therapies, Institute of Medical Science, Tokyo Medical University, Shinjuku, Tokyo 160-8402, Japan

**Keywords:** elementary school, kindergarten, alcohol use, parental factors, prevention programs

## Abstract

Parental experience of initiation of alcohol drinking has been identified as one of the early causes of alcohol drinking in preadolescents in many countries, including Japan. This study identified the association between parental alcohol-related knowledge and the initiation of alcohol use among preadolescent students in an urban area in Japan. Self-administrated questionnaires were distributed to 420 parents of kindergarteners and elementary school students, of which 339 were filled and returned (response rate: 81%). The parents’ experience in initiating alcohol drinking in their children and their knowledge about the effects of alcohol on youth were explored. The requirements for drinking prevention programs for youth were also investigated. The result showed that a significantly higher proportion of parents of elementary school students had experiences of initiating alcohol use in their children compared to parents of kindergarten children. The parents’ knowledge regarding the effects of alcohol on youth showed no significant difference between the two parent groups. These data indicate that the age of children is the only factor as opposed to parents’ knowledge. We also found significant differences in the requirements of prevention programs between the two parent groups. The results of this study can contribute to the design of alcohol prevention programs for these parents, which could reduce the onset of children’s drinking.

## 1. Introduction

Underage alcohol use is a serious problem associated with various health issues [1,2] and problem behaviors [3,4,5]. This problem is worldwide in its scope, although the prevalence varies across countries. In the United States, a high prevalence of alcohol intake among youth is reported by the National Survey Drug Use and Health [6]. In Asian countries, especially Thailand, Vietnam, and the Philippines, a high prevalence ratio of alcohol use has also been reported [7,8,9]. A similar prevalence rate has been reported in the urban areas of China within the past years [10]. Based on these reports, identifying students at risk of alcohol consumption [11], their parents’ alcohol-related knowledge, and the development of effective prevention methods are necessary to reduce the prevalence of underage alcohol use in Japan.

Alcohol misuse negatively affects young people and has become one of the leading causes of premature death [12]. The precise age of drinking onset is associated with subsequent alcohol dependence as well as various other health problems. For example, among persons in the U.S. who started drinking alcohol before 14 years, 13% became alcohol-dependent compared with only 2% of those who waited until they were 21 years or older [3,4]. Adolescent drinking does not only affect health but also causes maladaptive behavior, such as violence [13] and suicide [14]. Adolescents’ alcohol drinking, smoking, and drug-use behavior are interdependent in such a way that when a person starts drinking alcohol, he/she may likely start smoking and using drugs [15,16]. A survey on senior high school students revealed strong positive correlations between the initiation of alcohol drinking, smoking, and sexual intercourse [17]. The above findings indicate the need to identify factors associated with preadolescent alcohol drinking.

Given the negative health effects of alcohol drinking at younger ages, a large number of studies have been conducted on alcohol-related drinking among adolescents. For instance, several studies have sought to find out behavioral problems associated with the early onset of drinking such as whether students who began drinking alcohol at early adolescence frequently engaged in delinquency in later adolescence [18]. In addition, many studies have investigated the effects of individual and environmental factors on alcohol drinking, smoking, and drug use among high school students and adolescents [7,11,17,19,20,21,22,23,24,25]. However, only a few studies are available on alcohol use among preadolescents (e.g., elementary school students aged from 6 to 12 in Japan) [26], and fewer studies have been conducted on parents’ initiation of alcohol use among kindergarteners (age < 5) in Japan. This view is supported by the review of proceedings of the Research Society on Alcoholism meeting in 2003, which indicated that only a few studies are available on the onset of alcohol drinking among elementary students [26]. Stemmed from these findings, the current study, which seeks to investigate parents’ experience in initiating alcohol drinking and parents’ alcohol-related knowledge with a focus on elementary and kindergarten students, is deemed necessary.

Parental factors have repeatedly been identified as one of the risk factors for students’ alcohol use (reviewed in [27]). The positive link between maternal and paternal alcohol use behaviors was associated with alcohol drinking in adolescents [28]. Parents have also been identified as a source of alcoholic beverages for their children, which can increase the subsequent risky drinking (reviewed in [29]). The alcohol supply by parents, which includes using alcohol in the presence of the children, was associated with an increase in the number of cases of alcohol consumption in children and its impact was more significant on younger children, especially girls, compared to adolescents [30]. School information campaigns moderated the relationship between adolescent binge drinking and parental consumption [22], not peer consumption, indicating a likely high influence of parental alcohol consumption on students. Consequently, the prevention programs directed at parents might help to reduce alcohol drinking among children.

Although both preadolescent prevention efforts and parental education seem important for reducing adolescent drinking, there is no available study for such interventions in Japan. Based on the frequently reported association between parental behavior and children’s alcohol drinking, we hypothesized that alcohol supply by parents would initiate alcohol drinking in children. We also analyzed the relationship between parents’ knowledge of alcohol-related problems on youth and their alcohol supply. 

## 2. Materials and Methods

A total of 420 parents of children in one elementary school (6–12 years old) and three kindergartens (5–6 years old) in Yokohama, Japan, an urban area neighboring the Tokyo metropolitan area, were recruited to participate in the study. The parents of the students in the four schools were notified about the study through the teachers. To ensure anonymity and security, no online tools were used in this study. Thereafter, a paper-based self-administered questionnaire was delivered to the 420 parents of the students. Of this figure, 339 parents (*n* = 172 parents of kindergarteners and 167 parents of elementary students) returned the filled and completed questionnaires (response rate: 81%). The completed questionnaires were placed in an envelope to ensure anonymity along with parents’ written informed consent, and then parents were required to post them to boxes at these schools. The questionnaire sheet and written informed consent were anonymized. The data manager separated them and the data analyst analyzed the anonymized data. The study was approved by the Institutional Review Board of Yokohama Soei University (No. 27-010).

In the questionnaire, participants were asked to fill in the following information: parent’s sex and age (male/female; 20s, 30s, 40s, 50s, and 60s or more) and whether their children are in elementary school or kindergarten. The parents were asked a binary variable (yes/no) question concerning their provision of alcoholic beverages to their children. Since underage alcohol drinking is commonly known as illegal in Japan, to prevent parents’ concerns about disclosing this information, we did not provide the details regarding this question; that is, the information concerning the frequency and volume of alcohol beverages supplied to children was not included. The answers that enable us to identify the responder were also eliminated, as well as any other type of identifying information. 

Parents’ knowledge of the health effects of alcohol drinking on youth was determined using nine items. These items were selected from previous studies [31,32]. Participants were asked to check all that they believed to be true (i.e., multiple answers were allowed): (1) Drinking prevents appropriate physical growth and disturbs the homeostasis of sex hormones. (2) A large quantity of alcohol induces acute alcoholism. (3) Drinking induces brain malfunctioning (e.g., memory problems). (4) Drinking induces deterioration of the stomach lining. (5) Drinking frequently induces liver diseases such as hepatitis and cirrhosis. (6) Drinking induces pancreatic diseases such as pancreatitis and diabetes. (7) Excessive alcohol drinking sometimes induces death. (8) Alcohol drinking is addictive. (9) Alcoholism or alcohol addiction is associated with personality disorders.

Parents’ perceived requirements for alcohol prevention programs for elementary students were assessed using six items selected from a previous survey conducted in Japan [33]. Again, participants had to select all that they believed applied: (1) Lecture on the effects of drinking on the health of youths. (2) Lecture on the process of developing alcoholism. (3) Lecture on the laws regarding youth alcohol intake. (4) Presentations by alcoholics on their alcoholism experiences. (5) Investigation of the effects of drinking on the physical constitution. (6) Establishment of a consultation center on drinking.

We examined differences between the parents of elementary school students and kindergarten students on their knowledge of the effects of alcohol on youths and their perceived requirements for the alcohol prevention program. The chi-square test was used to assess age differences, and Fisher’s exact test was used for all other items, including responses for the main variables. Logistic regression analysis was used to calculate odds ratios and 95% confidential intervals (CIs) of odds ratios (OR) for all variables for parents of elementary school students compared to the parents of kindergarteners. *p*-values were adjusted by false discovery rate (Benjamini, Kreiger, and Yekutieli) to yield Q-values, considering the multiple independent tests. P-values were adjusted among the items of parents’ knowledge of the health effects of alcohol drinking and the items of requirements for the alcohol drinking prevention programs, respectively. All analyses were conducted using GraphPad Prism (ver. 7.0.03, GrapPad Software Inc. San Diego, CA, USA).

## 3. Results

Table 1 shows the parents’ general characteristics. The results showed no significant difference in sex (*p* = 0.93) between the groups. There was a significant age difference (*p* < 0.0001, chi-square test) between the groups: mode—the largest proportion of parents of kindergarteners and elementary school students was in their 30s (*n* = 107, 62.2%) and 40s (*n* = 111, 66.5%), respectively; thus, the latter group was generally older, as expected.

Table 2 summarizes participants’ responses for the main variables and compares them between the child groups. Only two parents of kindergarteners had experience in initiating drinking in their children (1.2%), whereas nearly all parents of elementary school students (*n* = 161, 96.4%) had such an experience. This difference was significant (*p* < 0.0001).

We observed no significant group differences in parents’ responses to the nine items evaluating their knowledge of the health effects of alcohol drinking on youth. The odds ratios (ORs) ranged from 0.80 to 1.6. Among the nine items, “(2) Large quantity of drinking induces acute alcoholism” garnered the highest responses among both parents of kindergarteners (*n* = 164, 94.2%) and parents of elementary school students (*n* = 160, 95.8%, OR = 0.71, 95% CI (confidence interval): 0.26–1.9). More than 80% of both groups reported knowledge of “(7) Excessive drinking sometimes induces death” (OR = 1.1, 95% CI: 0.62–2.1) and “(8) Drinking is addictive,” (OR = 0.91, 95% CI: 0.5–1.6), while less than 50% of these parents knew that “(4) Drinking induces deterioration of the stomach lining” (OR = 1.1, 95% CI: 0.73–1.7) and “(6) Drinking induces pancreatic disease such as pancreatitis and diabetes” (OR = 0.80, 95% CI: 0.52–1.2).

As for the six items concerning parents’ requirements for drinking prevention programs, we observed significant differences in four items between the parent groups: “(1) Lectures on effects of drinking on youth health” (*p* = 0.00090, Q = 0.0027, OR = 5.4, 95% CI: 1.8–16), “(3) Lectures on laws regarding youth drinking” (*p* < 0.0001, Q = 0.00060 OR = 2.9, 95% CI: 1.9–4.5), “(4) Presentations by alcoholics on their alcoholism experiences” (*p* = 0.0066, Q = 0.013, OR = 1.8, 95% CI: 1.2–2.8), and “(6) Establishment of a consultation center for drinking” (*p* = 0.020, Q = 0.030, OR = 2.0, 95% CI: 1.1–3.5). All these items showed higher demand by the parents of kindergarten children (OR values ≥ 1.0).

Table 3 shows a comparison of parents with or without the experience of initiating their children’s drinking. We observed no significant difference in sex (*p* = 0.86), but there was a significant age difference (*p* < 0.0001, chi-square test). Among parents who did not have the initiating experience, those in their 30s were the dominant age group (*n* = 106, 60.2%), while among those with such experience, those in their 40s were the dominant group. We observed no significant difference among the nine knowledge items. As for the requirements of a drinking prevention program, “(3) Prognosis of students with alcohol addiction” (*p* < 0.0001) showed the greatest difference; items (1), (4), and (6) also showed significant differences (*p* < 0.05 and Q < 0.05).

## 4. Discussion

Here, we analyzed the drinking initiation induced by parents for elementary school children and kindergarten and the association of this initiation with parental knowledge regarding youth alcohol drinking. One of the difficulties in this line of research is assuring the validity of data collected from such young students. Furthermore, research on adolescents has shown that parents are major alcohol suppliers [29]. Given this background, we designed a self-administrated questionnaire to understand parents’ knowledge of the health effects of alcohol and their experiences of initiating drinking in their children. We also analyzed parents’ requests to provide data for designing effective prevention programs.

The most profound difference between the parents of elementary school students and kindergarteners was their experience of initiating their children’s drinking; that is, very few parents of kindergarteners had such initiation experiences (1.16%) while nearly all parents of elementary school students had such experiences (96.41%). Meanwhile, no significant differences were observed in alcohol-related knowledge. Therefore, initiation experiences seem to be more dependent on children’s age than on parents’ knowledge. These initiation experiences were pervasive regardless of elementary school students’ age, so it seems that proceeding to elementary school (5 or 6 years old) is a clear borderline for initiation experiences.

A survey among Japanese students showed that students who initiated alcohol drinking and cigarette smoking at 12 years of age or less showed greater rates of dependency or addiction to these substances at 15 years of age or older [24]. As noted earlier, 96.4% of the parents of elementary school students had the experience of initiating alcohol drinking in their children. Parental communication and socialization can influence children’s susceptibility to alcohol use initiation; that is, talking with elementary school children about harmful consequences of alcohol use reduced the children’s susceptibilities in the family where the parent drank frequently [34]. Independent reports collected from the father, mother, and children each revealed that maternal alcohol abuse was associated with attention and conduct problems observed in adolescent children [35]. These findings indicated the importance of appropriate education on alcohol drinking among parents of students to prevent drinking. Notably, however, we found no difference in parents’ knowledge between the groups. In Japan, sake (rice wine) has been considered as not just alcohol but also a mysterious beverage and used for various purposes, including, for example, the agreement of pledge of brothers between leaders of different groups by their simultaneous drinking [36]. Especially in the region of Shinto, sake is supplied in various situations, such as shrine festivals and wedding ceremonies [37]. These might weaken the perceived health risks of alcohol for both parents and children. Furthermore, since the initiation was caused by parents, children were given access to alcohol without a sense of immorality. This situation is rather different from parental problem drinking [27,34,35,38,39,40]. From our result, a high proportion of parents of elementary school students have experience of providing alcohol to their offspring, which indicates that the education of parents is essential. It might be necessary to investigate more carefully the communication strategies between parents and children, which were identified as influential factors for children’s alcohol-drinking behaviors [34].

It is increasingly difficult to obtain accurate data from elementary school students and kindergarteners using self-reported questionnaires because of their limited ability to respond to the questions. Therefore, we analyzed parents’ data instead. We found that parents’ knowledge did not differ according to their experience of initiating alcohol drinking in children. One possible reason is that the data were collected in an urban area close to Tokyo, where most of the residents have similar and high levels of education and economic backgrounds. The current results might not be generalizable to the provincial area, including families with more heterogeneous backgrounds. In addition, based on the extremely high occurrence of parents of elementary school students providing alcoholic beverages to them, the parents without this experience might not be concerned with legal limitations but instead may consider the impacts of drinking on their children’s health; in other words, these parents might regard their children who grew up as elementary school students to be strong enough to resist the drinking of alcohol.

Participants’ requirements for drinking prevention programs differed according to their children’s school. We might consider their required contents as parents’ demands for specific knowledge. Thus, the selected items could contribute to the design of parental education programs, which could help to reduce the initiation of children’s alcohol usage. While school programs are known to be more effective for managing problematic behaviors among students in Japan (e.g., drinking prevention) when compared to neighborhood programs, the parents were not included in the comparisons [41]. However, the importance of parental education was not evaluated in our study.

This study has several limitations. First, we collected data only in urban areas of Japan, and the sample size was small. In Japan, the constitutive members of a family and cultural events, such as festivals and ceremonial occasions, differ widely between urban and rural areas, and these factors might affect the initiation of alcohol drinking. A larger sample collected from various regions of Japan (both urban and rural) is necessary to evaluate the generalizability of our findings. Second, we hypothesized that parental knowledge would influence the initiation of alcohol drinking and administered the questionnaire only to parents of elementary school students and kindergarteners. Therefore, other factors that affect the initiation of alcohol drinking were not explored. Thirdly, all questionnaires were designed as binary questions, with no dummy or test questions. One of the major problems of this study is that parents are the source of the information even though the questions include inducing their children to illegal alcohol drinking. The quantitative value (e.g., the amount of alcohol supply) was not determined. The different effect of parental and maternal alcohol use on children was reported; for example, paternal drinking showed a higher association of male adolescents rather than maternal drinking [42]. That difference was not found in female adolescents, and this study also showed a high link of peer drinking [42]. Our data included fewer male parents, which possibly causes a bias of the observed association. We were concerned about the low response of this study since the alcohol supply is an explicitly problematic behavior to induce illegal drinking. Thus, we did not include the questions to provide detailed family attributes. However, this simplified questionnaire limits the relative relationship of the other factors to induce children’s alcohol drinking.

## 5. Conclusions

In conclusion, this study investigated the differences in parents’ experience of initiating alcohol use in their children. Parents of elementary school children, in general, had experience in initiating alcohol drinking in their children, whereas parents of kindergartners rarely had such an experience. We also investigated young people’s knowledge of alcohol drinking and found no significant difference between the two parent groups, suggesting that the age of the children is the only factor. The elementary school students reported a significantly greater number of requirements for drinking prevention programs. Taken together, effective education, especially for parents of elementary school children, is necessary.

## Figures and Tables

**Table 1 children-08-00258-t001:** Characteristics of participants.

Characteristic		Kindergarten		Elementary School		*p*-Value
		*n*	(%)	*n*	(%)	
Total	-	172	-	167	-	
Sex	Male	18	(10)	18	(11)	0.93
	Female	154	(90)	149	(89)	
Age	20s	15	(8.72)	6	(3.59)	<0.0001 ***
	30s	107	(62.2)	42	(25.1)	
	40s	45	(26.2)	111	(66.5)	
	50s	1	(0.581)	7	(4.19)	
	60s or more	0	(0.00)	1	(0.599)	

Note: *** *p* < 0.001.

**Table 2 children-08-00258-t002:** Comparison of parents of kindergarten and elementary school.

Item		Kindergarten *n* (%)	Elementary School *n* (%)	*p*-Value (Q-Value)	Odds Ratio (95% CI)
Total		172	167		
Parental experience of initiation of drinking use in children					
Yes		2 (1.20)	161 (96.4)	<0.0001	2281 (453.6– 11,470)
No		170 (98.9)	6 (3.60)		
Knowledge of the possible effects of youth drinking					
1	Drinking prevents physical growth and disturbs the homeostasis of sex hormones	140 (81.4)	126 (75.4)	0.19 (0.80)	1.4 (0.85–2.4)
2	A large quantity of drinking induces acute alcoholism	162 (94.2)	160 (95.8)	0.62 (0.80)	0.71 (0.26–1.9)
3	Drinking induces brain malfunctioning (e.g., memory problems)	129 (75.0)	128 (76.6)	0.80 (0.80)	0.91 (0.56–1.5)
4	Drinking induces deterioration of the stomach lining	87 (50.6)	80 (47.9)	0.66 (0.80)	1.1 (0.73–1.7)
5	Drinking induces liver diseases such as hepatitis and cirrhosis	135 (78.5)	117 (70.1)	0.080 (0.72)	1.6 (0.95–2.6)
6	Drinking induces pancreatic diseases such as pancreatitis and diabetes	73 (42.4)	80 (47.9)	0.33 (0.80)	0.80 (0.52–1.2)
7	Excessive drinking sometimes induces death	148 (86.0)	141 (84.4)	0.76 (0.80)	1.1 (0.62–2.1)
8	Drinking is addictive	144 (83.7)	142 (85.0)	0.77 (0.80)	0.91 (0.50–1.6)
9	Alcoholism disturbs personality	118 (68.6)	118 (70.7)	0.72 (0.80)	0.91 (0.57–1.4)
Requirements of the drinking prevention program					
1	Lectures on effects of drinking on youth health	168 (97.7)	148 (88.6)	0.00090 *** (0.0027 **)	5.4 (1.8–16)
2	Lectures on process of alcoholism development	109 (63.4)	98 (58.7)	0.44 (0.44)	1.2 (0.79–1.9)
3	Lectures on laws regarding youth drinking	103 (59.9)	57 (34.1)	<0.0001 *** (0.00060 ***)	2.9 (1.9–4.5)
4	Presentations by alcoholics on their alcoholism experiences	97 (56.4)	69 (41.3)	0.0066 ** (0.013 *)	1.8 (1.2–2.8)
5	Investigation on the effects of drinking on physical constitution	80 (46.5)	69 (41.3)	0.38 (0.44)	1.2 (0.8–1.9)
6	Establishment of a consultation center on drinking	40 (23.3)	22 (13.2)	0.020 * (0.030 *)	2.0 (1.1–3.5)

Note, CI: confidence interval of the odds ratio. * *p* < 0.05, ** *p* < 0.01, and *** *p* < 0.001 for both *p*-values and Q-values.

**Table 3 children-08-00258-t003:** Comparison of parents with/without the experience of initiating alcohol drinking in their children.

Item		With Experience *n* (%)	Without Experience *n* (%)	*p*-Value (Q-Value)	Odds Ratio (95% CI)
Total		176	1163		
Knowledge of possible effects of youth drinking					
1	Drinking prevents physical growth and disturbs the homeostasis of sex hormones	142 (80.7)	124 (76.1)	0.30 (0.80)	1.30 (0.78–2.2)
2	Large quantity of drinking induces acute alcoholism	165 (93.8)	157 (96.3)	0.28 (0.80)	0.57 (0.21–1.6)
3	Drinking induces brain malfunctioning (e.g., memory problems)	131 (74.4)	126 (77.3)	0.54 (0.80)	0.85 (0.52–1.4)
4	Drinking induces deterioration of the stomach lining	87 (49.4)	80 (49.1)	0.95 (0.80)	1.0 (0.66–1.6)
**5**	Drinking induces liver diseases such as hepatitis and cirrhosis	136 (77.3)	116 (71.2)	0.20 (0.72)	1.4 (0.84–2.2)
6	Drinking induces pancreatic diseases such as pancreatitis and diabetes	73 (41.5)	80 (49.1)	0.16 (0.80)	0.74 (0.48–1.1)
7	Excessive drinking sometimes induces death	149 (84.7)	140 (85.9)	0.75 (0.80)	0.91 (0.5–1.7)
8	Drinking is addictive	147 (83.5)	139 (85.3)	0.66 (0.80)	0.87 (0.49–1.6)
9	Alcoholism disturbs personality	121 (68.8)	115 (70.6)	0.72 (0.80)	0.92 (0.58–1.5)
Requirements of the drinking prevention program					
1	Lecture on effects of drinking on youth health	169 (96.0)	147 (90.2)	0.033 * (0.049 *)	2.6 (1.1–6.6)
2	Lecture on process of alcoholism development	110 (62.5)	97 (59.5)	0.57 (0.57)	1.1 (0.73–1.8)
3	Lecture on laws regarding youth drinking	103 (58.5)	57 (35.0)	<0.0001 (0.00060)	2.6 (1.7–4.1)
4	Presentations by alcoholics on their alcoholism experiences	98 (55.7)	68 (41.7)	0.010 ** (0.031 *)	1.755 (1.1–2.7)
5	Investigation on the effects of drinking on physical constitution	81 (46.0)	68 (41.7)	0.42 (0.51)	1.2 (0.77–1.8)
6	Establishment of a consultation center on drinking	40 (22.7)	22 (13.5)	0.028 * (0.049 *)	1.9 (1.1–3.3)

Note: * *p* < 0.05 and ** *p* < 0.01 for both *p*-values and Q-values.

## Data Availability

The data that support the findings of this study are available from the corresponding author, M.S., upon reasonable request.

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
