# Peer review of "Parents’ Initiation of Alcohol Drinking among Elementary and Kindergarten Students"

_children, 2021, doi:10.3390/children8040258_

Round 1

Reviewer 1 Report

Eto and Masahiro explore differences between parents of kindergarten and elementary school children in the initiation of alcohol to their children. They also explore parent’s knowledge of the effects of alcohol and the needs for prevention programs. Although interesting, I have the following concerns:

The authors attempt to generalise their findings despite the sample being relatively small and confined to an area with very specific demographics. Whether these findings are representative of the wider population is difficult to assess and more data from diverse neighbourhoods would be valuable. I am also concerned about the small number of male participants. Males are inherently less risk adverse than females and therefore they might be more likely to initiate alcohol than females. However, the small numbers prevent any meaningful analysis stratified by sex.

 The authors use complete pooling for the data from the kindergarten responses. This assumes that there is no effect of class/school on the outcomes, this is an assumption which needs to be explored to ensure that the variance within the dataset is appropriately captured and modelled. Furthermore, the authors have not considered the impact of multiple testing on their findings, potentially inflating type I error rates.

The introduction is lengthy and could be shortened without loss of emphasis. It currently feels more like a literature review.

There is no information on the response rate (i.e. how many questionnaires were distributed)

There appears to be an error in Table 2. Data for elementary school has the n repeated in parentheses rather than %.

There are some minor typographical errors and several sentences are protracted, which causes uncertainty for the reader.

Author Response

Eto and Masahiro explore differences between parents of kindergarten and elementary school children in the initiation of alcohol to their children. They also explore parent’s knowledge of the effects of alcohol and the needs for prevention programs. Although interesting, I have the following concerns:

The authors attempt to generalise their findings despite the sample being relatively small and confined to an area with very specific demographics. Whether these findings are representative of the wider population is difficult to assess and more data from diverse neighbourhoods would be valuable. I am also concerned about the small number of male participants. Males are inherently less risk adverse than females and therefore they might be more likely to initiate alcohol than females. However, the small numbers prevent any meaningful analysis stratified by sex.

The authors use complete pooling for the data from the kindergarten responses. This assumes that there is no effect of class/school on the outcomes, this is an assumption which needs to be explored to ensure that the variance within the dataset is appropriately captured and modelled. Furthermore, the authors have not considered the impact of multiple testing on their findings, potentially inflating type I error rates.

We thank the reviewer for this pertinent observation. Following the reviewer’s comment, the current manuscript has been edited by a professional English editor. Also, we have carefully revised the manuscript to reflect the reviewers suggestions.

The introduction is lengthy and could be shortened without loss of emphasis. It currently feels more like a literature review.

 We have shortened the Introduction section as suggested. See pages 2-3.

There is no information on the response rate (i.e. how many questionnaires were distributed)

The response rate has been added in the Abstract section and Materials and Methods section as follows:

Self-administrated questionnaires were distributed to 420 parents of kindergarteners and elementary school students, of which 339 were filled and returned (response rate: 81%).

There appears to be an error in Table 2. Data for elementary school has the n repeated in parentheses rather than %.

We thank the reviewer for this observation. The error has been corrected.

There are some minor typographical errors and several sentences are protracted, which causes uncertainty for the reader.

We thank the reviewer for pointing this out. We have corrected typographical errors, and the manuscript has been edited by a professional English language editor.

Reviewer 2 Report

Thank you for providing me with the opportunity to review this paper.

This is a potentially interesting article looking at Parents’ initiation of alcohol drinking among elementary school and kindergarten students. I don’t think that this paper makes a large contribution to the research area in its current form, but it is a relatively important topic with much promise.

Unfortunately, the manuscript in its current form and the way that the research has been conducted does need quite a bit of work before I would recommend publication.  The main issues are that the findings do not appear to make a novel contribution to the area, there are questions about the methods, with poor framing in terms of the wider literature and field. Throughout the manuscript there are grammatical and structural issues. I have included my comments below and hope the authors find these comments useful.

Abstract: This is mostly clear, although grammatical issues are throughout. The aims of the research are not particular clear and this section does not really highlight the importance of this topic for the research area.

  • ‘parent supply’? What does this mean? Parents who supply children with alcohol?
  • Some grammatical issues in the first sentence (“one trigger which may start alcohol drinking in preadolescence?”
  • Some long sentences would benefit from punctuation
  • ‘certain requirements of prevention programmes’ – this is unclear as to what it refers to?
  • ‘This data’ not ‘these data’?

Introduction: This section is relatively well-written, with some critical points. However, the structure could be improved to aid readability and there are a number of grammatical issues. A particular focus should be on the clarity and strength of the rationale, which does not follow very clearly from the presented introduction.

  • Is mostly well-structured, with some critical points. However, needs a bit of work throughout to further clarify the structure and strengthen the rationale.
  • Could include more recent research (up to 90s? focuses on surveys in 1996? 2008?), lots have been published in last 5-10 years in this area.
  • Some typos “experience” “Viet Nam”?
  • It is good to see the authors contextualise drinking in the country research conducted in.
  • 7th grade – would be good to define for those who do not know age etc.
  • Begin preventative efforts even earlier – depends on the preventative efforts? Which type?
  • Line 43 ‘has’ become one of the leading causes not ‘have’.
  • Line 50 – illegal or illicit drug use? Key differences.
  • Some very long sentences which require more punctuation.
  • Preadolescent (<13) drinking – line 63, can you provide examples of validity/ reliability problems? Is it that there is little research in this area?
  • Line 68 “became carrying guns” does not make sense.
  • Line 69 and 70 – grammatical issues.
  • Argument not always clear of focused throughout.
  • Line 77 – contributing individual and environmental factors – not sure how cigarettes and drugs link here? How are cigarettes and drugs individual or environmental factors? Do they mean poly-drug use?
  • Line 80 is key here – limited research exists, but the authors make the connection between this and effective prevention programmes. How are these effective programmes? Do we know if these are needed if the research is limited? Some parts here are unclear.
  • This section does jump between prevalence data and prevention information and could do with a clearer structure in places. There is not one clear theory and evidence is not always presented all at once for each approach described – for example, a distinction is made in line 85 onwards about contextual and individual/ interpersonal factors, but throughout these are referred to in different area or ways. Line 96 then ends (after discussing access) to focus on knowledge and education providing approaches, before going back to parental problems in line 98 and line 111 talking about parental education measures.
  • Line 115 – “the children” – which children?
  • The rationale could be much stronger at the end of the introduction to introduce the current study. A summary of many different areas is provided, but this does not lead clearly into the rationale. Research questions/ hypotheses are brief and could be better formulated/ presented.

Materials and methods: While most of the information here is presented, I still have some concerns about ethical processes and would question why measures were designed when valid measures exist. I do think some of the findings are linked to the measures used, which were not validated.

  • “High price” land could be better phrased? There is a literature on SES/ deprivation, yet the introduction does not cover this?
  • School teachers introduced the study to parents – could there have been a power issue/ coercion with parents compelled to ensure participants took part? Ethically could parents decline taking part? When were teachers in touch with students?
  • Unclear – elementary students gave parents the questionnaire, yet it was handed to kindergarten parents? Could children have completed the questionnaires? Could these not have been distributed online through school emails?
  • Completed questionnaires stored with consent forms – were these immediately separated to ensure anonymity? Otherwise this could be ethically problematic.
  • I’m not certain how age data was collected, or how the question about providing alcoholic beverages to children was worded. For example, did it state where and when? It is understandable that this information may be illegal, but the question may not have been understood – for example, a sip or a drink provided. Social desireability?
  • Examples of questions or references of validated items would be appropriate here.
  • Some of the words/ phrases/ jargon used in questions may not have been understood by parents. For example, there is no definition provided of acute alcoholism or homeostasis. Also ‘chugging’?
  • It is not clear what parents were asked to assess in terms of the drinking prevention programme – is this what they think would be beneficial to them? For their child? Some of these options do not seem appropriate for prevention programmes and children of that young age? Why were these chosen and others not?
  • Please clarify differences in age between parents of elementary and kindergarten children.
  • There should have been some dummy/ test questions in this, as many answers provided are likely to be answered in a certain way, so it will be unlikely differences will be found. Binary responses are often problematic.

Results: While most information is included in this section, it is not always clear to understand what was found. Reporting of findings could be improved.

  • 95% confidence intervals – on what? Errors?
  • Usual statistics are reported to 2dp., please check journal requirements.
  • Very simplistic analyses.
  • No differences found between kindergarten and elementary school – very similar opinions (largely due to the design of the measures?)
  • Why was more information not collected? Qualitative research could have focused on how/ why this initiation was completed? Numbers do not say much about actual experiences?
  • No focus on validating the questionnaire/ measures used?
  • While amounts were not asked due to the illegal nature of these behaviours, there is little indication of how this question is understood – looking at the parent’s own consumption/ attitudes towards alcohol might have been more worthwhile? As well as potentially using short, validated existing measures such as the AUDIT, for example?
  • The knowledge questions are quite short and coverage is limited – I’m not sure about the validity of these questions in focusing on knowledge (as mentioned before above).

Discussion: This seems quite brief compared to the introduction and does not seem to provide a satisfactory overview of the topic area. The novel contribution to the topic area is not clear, the implications of findings are not stated and provided limitations are quite generic, with no reasonable suggestions for future research. More should be done to situate findings in the wider field.

  • Focus on parents, when actually peers may be a bigger influence on drinking behaviour?
  • A positive is provided that a focus is on parent’s knowledge, yet the questionnaires were distributed through students.
  • It is likely the findings are not generalisable beyond the immediate schools.
  • While some discussion of findings is provided, it does not add too much to what is already known. Some aspects are unclear or do not follow from findings clearly.
  • The structure again of this section could be improved.
  • Education is seen as a key finding, but there is lots of work in this area showing how education/ changing attitudes has low efficacy. A focus could have instead been on what is provided at the schools in question and then how what is being recommended differs from this.
  • Limitations – small sample size, design of measures, knowledge questions were limited. Other factors of drinking initiation were not explored.

Author Response

Reviewer 2

Thank you for providing me with the opportunity to review this paper.

This is a potentially interesting article looking at Parents’ initiation of alcohol drinking among elementary school and kindergarten students. I don’t think that this paper makes a large contribution to the research area in its current form, but it is a relatively important topic with much promise.

Unfortunately, the manuscript in its current form and the way that the research has been conducted does need quite a bit of work before I would recommend publication.  The main issues are that the findings do not appear to make a novel contribution to the area, there are questions about the methods, with poor framing in terms of the wider literature and field. Throughout the manuscript there are grammatical and structural issues. I have included my comments below and hope the authors find these comments useful.

We appreciate the reviewer’s valuable observations and suggestions. We have revised the manuscript to address the issues pointed out by the reviewer. The novel contributions of the manuscript to the research area has been added in the Introduction sections. The methods section has been improved by addiding the number of participants, procedure for recruiting the participants, the return rate and contents of the questionnaire. The current manuscript has been edited by a professional English editor.

Abstract: This is mostly clear, although grammatical issues are throughout. The aims of the research are not particular clear and this section does not really highlight the importance of this topic for the research area.

To address the reviewer’s comment, we have clearly added the aims of the research and relevance of the study to the research area in the Abstract section.  The grammatical issues have been addressed with the assistance of a professional English editor.

‘parent supply’? What does this mean? Parents who supply children with alcohol?

Some grammatical issues in the first sentence (“one trigger which may start alcohol drinking in preadolescence?”

In line with the reviewer’s comments, the awkward words and phrases in the abstract has been revised.

Some long sentences would benefit from punctuation

To address the reviewer’s comments, we have used punctuation to separate long sentences.

‘certain requirements of prevention programmes’ – this is unclear as to what it refers to?

To address the reviewer’s comment, we have further described the programs.

‘This data’ not ‘these data’?

 We have revised this part.

Introduction: This section is relatively well-written, with some critical points. However, the structure could be improved to aid readability and there are a number of grammatical issues. A particular focus should be on the clarity and strength of the rationale, which does not follow very clearly from the presented introduction.

Is mostly well-structured, with some critical points. However, needs a bit of work throughout to further clarify the structure and strengthen the rationale.

Could include more recent research (up to 90s? focuses on surveys in 1996? 2008?), lots have been published in last 5-10 years in this area.

To address the reviewer’s comments, we have deleted outdated citation and cited the recent researches instead. Most part of the manuscript has been revised to improve clarity and structure.

Some typos “experience” “Viet Nam”?

We thank the reviewer for this observation. We have corrected the typographical errors.

It is good to see the authors contextualise drinking in the country research conducted in.

Thank you for the positive feedback. In accordance with the reviewer’s comments, the second paragraph of the Introduction was revised to describe the contextual analyses, e.g., references 13 and 14.

7th grade – would be good to define for those who do not know age etc.

According to the reviewer’s comments, we have removed these expressions from the second paragraph of the Introduction.

Begin preventative efforts even earlier – depends on the preventative efforts? Which type?

According to the reviewer’s comments, we have removed these expressions from the Introduction.

Line 43 ‘has’ become one of the leading causes not ‘have’.

We thank the reviewer for this observation. We have corrected this error.

Line 50 – illegal or illicit drug use? Key differences.

According to the reviewer’s comments, we have removed the sentence containing this expression.

Some very long sentences which require more punctuation.

To address the reviewer’s comments, we have split long sentences into shorter ones.

Preadolescent (<13) drinking – line 63, can you provide examples of validity/ reliability problems? Is it that there is little research in this area?

We thank the reviewer for this observation. We have revised the sentence to indicate the small survey on elementary students.

Line 68 “became carrying guns” does not make sense.

In line with the reviewer’s comment, we have deleted the sentence.

Line 69 and 70 – grammatical issues.

Argument not always clear of focused throughout.

We have revised this sentence to reflect the reviewer’s suggestion.

Line 77 – contributing individual and environmental factors – not sure how cigarettes and drugs link here? How are cigarettes and drugs individual or environmental factors? Do they mean poly-drug use?

We thank the reviewer for this insightful  observation. We have revised the sentence in the third paragraph of the Introduction.

Line 80 is key here – limited research exists, but the authors make the connection between this and effective prevention programmes. How are these effective programmes? Do we know if these are needed if the research is limited? Some parts here are unclear.

We thank the reviewer for these valuable comments. We have revised the fourth paragraph of the Introduction to indicate the connection between the needs for the development of effective prevention programs and the identification of factors that induce alcohol drinking in preadolescents.

This section does jump between prevalence data and prevention information and could do with a clearer structure in places. There is not one clear theory and evidence is not always presented all at once for each approach described – for example, a distinction is made in line 85 onwards about contextual and individual/ interpersonal factors, but throughout these are referred to in different area or ways. Line 96 then ends (after discussing access) to focus on knowledge and education providing approaches, before going back to parental problems in line 98 and line 111 talking about parental education measures.

We thank the reviewer for this insightful observation. We have revised the parts pointed out by the reviewer so the text reads more clearly and logically. As pointed by the reviewer, there was a section jump in the Introduction. In the revised manuscript, we removed this paragraph and revised the fourth and fifth paragraphs of the Introduction to fill in the gap. In the fifth paragraph, several less relevant sentences were eliminated.

Line 115 – “the children” – which children?

Thank you for pointing this out. This part has been revised to be clearer.

The rationale could be much stronger at the end of the introduction to introduce the current study. A summary of many different areas is provided, but this does not lead clearly into the rationale. Research questions/ hypotheses are brief and could be better formulated/ presented.

We have revised this part to clarify the rationale of the study.

Materials and methods: While most of the information here is presented, I still have some concerns about ethical processes and would question why measures were designed when valid measures exist. I do think some of the findings are linked to the measures used, which were not validated.

“High price” land could be better phrased? There is a literature on SES/ deprivation, yet the introduction does not cover this?

We have revised this part in line with the reviewer’s suggestion.

School teachers introduced the study to parents – could there have been a power issue/ coercion with parents compelled to ensure participants took part? Ethically could parents decline taking part? When were teachers in touch with students?

Unclear – elementary students gave parents the questionnaire, yet it was handed to kindergarten parents? Could children have completed the questionnaires? Could these not have been distributed online through school emails?

Following the points raised by the reviewer, we have revised this part so that it reads more clearly.

Completed questionnaires stored with consent forms – were these immediately separated to ensure anonymity? Otherwise this could be ethically problematic.

I’m not certain how age data was collected, or how the question about providing alcoholic beverages to children was worded. For example, did it state where and when? It is understandable that this information may be illegal, but the question may not have been understood – for example, a sip or a drink provided. Social desireability?

According to the reviewer’s comments, we have revised the first and second paragraphs of Materials and Methods to clarify the procedure of data collection.

Examples of questions or references of validated items would be appropriate here.

Some of the words/ phrases/ jargon used in questions may not have been understood by parents. For example, there is no definition provided of acute alcoholism or homeostasis. Also ‘chugging’?

 Difficult words/jargon such as “chugging,” which parents may find difficult to understand, have been replaced, while common/simple words have been retained.

It is not clear what parents were asked to assess in terms of the drinking prevention programme – is this what they think would be beneficial to them? For their child? Some of these options do not seem appropriate for prevention programmes and children of that young age? Why were these chosen and others not?

The prevention program as listed in the Table was aimed to guide parents to know more about how to raise their children so that the children are not induced to consume alcohol at an early age.

Please clarify differences in age between parents of elementary and kindergarten children.

We have clarified the differences in age between the two parents’ groups in the Result section as follows:

The result showed no significant difference in sex (P = 0.9254) between the groups, there was a a significant age difference (P < 0.0001, chi-square test) between the groups: mode—the largest proportion of parents of kindergarteners and elementary school students were 30s (n = 107, 62.2%) and 40s (n = 111, 66.5%), respectively;

There should have been some dummy/ test questions in this, as many answers provided are likely to be answered in a certain way, so it will be unlikely differences will be found. Binary responses are often problematic.

This has been stated as one of the limitations of the study.

Results: While most information is included in this section, it is not always clear to understand what was found. Reporting of findings could be improved.

95% confidence intervals – on what? Errors?

According to the reviewer’s comments, we have added the explanation “confidence intervals on odds ratio” in the footnote to Table 1. We also revised the third paragraph of the Result for clarity.

Usual statistics are reported to 2dp., please check journal requirements.

Very simplistic analyses.

We have reported the statistics following the journal’s requirements.

No differences found between kindergarten and elementary school – very similar opinions (largely due to the design of the measures?)

We have explained the possible reasons in the sixth paragraphn in the Discussions.

Why was more information not collected? Qualitative research could have focused on how/ why this initiation was completed? Numbers do not say much about actual experiences?

These issues have been reflected in the Limitation section.

No focus on validating the questionnaire/ measures used?

According to the reviewer’s comments, we have revised the third and fourth paragraphs of Material and Methods to indicate the resources used for the questionnaire.

While amounts were not asked due to the illegal nature of these behaviours, there is little indication of how this question is understood – looking at the parent’s own consumption/ attitudes towards alcohol might have been more worthwhile? As well as potentially using short, validated existing measures such as the AUDIT, for example?

The knowledge questions are quite short and coverage is limited – I’m not sure about the validity of these questions in focusing on knowledge (as mentioned before above).

To address these issues raised by the reviewer, we have revised the Materials and Methods section to accommodate part of the reviewer’s suggestions. We have also included parts of the reviewer’s concern in the limitation section.

Discussion: This seems quite brief compared to the introduction and does not seem to provide a satisfactory overview of the topic area. The novel contribution to the topic area is not clear, the implications of findings are not stated and provided limitations are quite generic, with no reasonable suggestions for future research. More should be done to situate findings in the wider field.

In line with the reviewer’s suggestion, we have revised the Discussion section to accommodate suggestions raised by the reviewer.

Focus on parents, when actually peers may be a bigger influence on drinking behaviour?

Considering the age group (Elementary and Kindergarten students), we felt that they would be more attached to their parents than peers. Also, a large number of studies have been conducted to investigate peer group influence on drinking behavior, unlike parental influence, which seems to be underexplored.

A positive is provided that a focus is on parent’s knowledge, yet the questionnaires were distributed through students.

We thank this observation. We revised the Material and Methods.

It is likely the findings are not generalisable beyond the immediate schools.

We have reflected this in the limitations as follows: This study has several limitations. First, we collected data only in urban areas of Japan, and the sample size was small. In Japan, the constitutive members of a family and cultural events, such as festivals and ceremonial occasions, differ widely between urban and rural areas, and these factors might affect the initiation of alcohol drinking. A larger sample collected from various regions of Japan (both urban and rural) is necessary to evaluate the generalizability of our findings.

While some discussion of findings is provided, it does not add too much to what is already known. Some aspects are unclear or do not follow from findings clearly.

We have revised the discussion section for clarity.

The structure again of this section could be improved.

Education is seen as a key finding, but there is lots of work in this area showing how education/ changing attitudes has low efficacy. A focus could have instead been on what is provided at the schools in question and then how what is being recommended differs from this.

This has been included as part of the limitations of this study.

Limitations – small sample size, design of measures, knowledge questions were limited. Other factors of drinking initiation were not explored.

This has been included as part of the limitations of this study.

Round 2

Reviewer 1 Report

The authors have used the previous comments to make improvements to the paper. There are still some typographical errors that need to be addressed and I suggest a good sweep of the paper. It is also my feeling that the introduction could be further shortened.

Table 2: Present real number rather than scientific value for OR. I also suggest that values (e.g. P and Q values) have a standardised number of decimal places.

Author Response

Comment 1: The authors have used the previous comments to make improvements to the paper. There are still some typographical errors that need to be addressed and I suggest a good sweep of the paper. It is also my feeling that the introduction could be further shortened.

Response 1: Thank you for this observation. According to your comment, the second paragraph of the Introduction has been eliminated. Further, several sentences in the other paragraphs of the Introduction have been shortened. We also asked a professional English proofreader to edit the latest version of the manuscript and we carefully revised the manuscript to eliminate remaining grammatical errors.

Comment 2: Table 2: Present real number rather than scientific value for OR. I also suggest that values (e.g. P and Q values) have a standardised number of decimal places.

Response 2: According to your comment, the OR values are now presented as real numbers. The P and Q values and percentages are expressed as standardized numbers.

Reviewer 2 Report

Thank you to the authors for making many of the proposed suggestions for the paper. I still think that the paper could be more concise throughout and better state the findings and original contribution to the research area. I still detect a number of grammatical/ typographical issues which should not be present in a published paper. The paper stills needs some restructuring and amending before being considered for publication.

Author Response

Comment 1: Thank you to the authors for making many of the proposed suggestions for the paper. I still think that the paper could be more concise throughout and better state the findings and original contribution to the research area. I still detect a number of grammatical/ typographical issues which should not be present in a published paper. The paper stills needs some restructuring and amending before being considered for publication.

Response 1: Thank you for these comments. According to your comments, we have revised the Abstract and last paragraph of the Discussion to clearly express the conclusions of this study. We have also shortened the Introduction (e.g., the second paragraph has been eliminated). Finally, we asked a professional English proofreader to edit the latest version of the manuscript and we have carefully revised the manuscript to eliminate remaining grammatical errors.